



# Mineral dust modelling with MADE3 in EMAC v2.54

Christof G. Beer[1], Johannes Hendricks[1], Mattia Righi[1], Bernd Heinold[2], Ina Tegen[2], Silke Groß[1], Daniel Sauer[1,4], Adrian Walser[3,4,1], and Bernadett Weinzierl[3]

[1]Deutsches Zentrum für Luft- und Raumfahrt (DLR), Institut für Physik der Atmosphäre, Oberpfaffenhofen, Germany
[2]Leibniz Institute for Tropospheric Research, Leipzig, Germany
[3]University of Vienna, Faculty of Physics, Aerosol Physics and Environmental Physics, Vienna, Austria
[4]Ludwig-Maximilians-Universität München, Meteorologisches Institut, Munich, Germany

**Correspondence:** Christof Beer (christof.beer@dlr.de)

**Abstract.** Mineral dust particles play an important role in the climate system, by e.g. interacting with solar and terrestrial radiation or facilitating the formation of cloud droplets. Additionally, dust particles can act as very efficient ice nuclei in cirrus clouds. Many Global Chemistry Climate Models (GCCMs) use prescribed monthly mean mineral dust emissions representative of a specific year, based on a climatology. It was hypothesized that using dust emission climatologies may lead to misrepresenta-

tions of strong dust burst episodes, resulting in a negative bias of model dust concentrations compared to observations for these episodes. Here, we apply the aerosol microphysics submodel MADE3 (Modal Aerosol Dynamics model for Europe, adapted for global applications, third generation) as part of the ECHAM/MESSy Atmospheric Chemistry (EMAC) general circulation model. We employ two different representations of mineral dust for our model simulations: i) a prescribed monthly-mean climatology of dust emissions representative of the year 2000; ii) an online dust parametrization which calculates wind-driven

mineral dust emissions at every model time-step. We evaluate model results for these two dust representations by comparison with observations of aerosol optical depth from ground-based station data. The model results show a better agreement with the observations for strong dust burst events when using the online dust representation compared to the prescribed dust emissions setup. Furthermore, we analyse the effect of increasing the vertical and horizontal model resolution on mineral dust properties in our model. The model is evaluated against airborne in situ measurements performed during the SALTRACE mineral

dust campaign (Saharan Aerosol Long-range Transport and Aerosol-Cloud Interaction Experiment, June/July 2013), i.e. observations of dust transported from the Sahara to the Caribbean. Results show that an increased horizontal and vertical model resolution is able to better represent the spatial distribution of airborne mineral dust, especially in the upper troposphere (above 400 hPa). Additionally, we analyse the effect of varying assumptions for the size distribution of emitted dust. The results of this study will help to identify the model setup best suited for future studies and to further improve the representation of mineral

dust particles in EMAC-MADE3.

## 1 Introduction

Mineral dust particles can influence the climate system in various ways. Atmospheric dust aerosols interact with solar and terrestrial radiation through absorption and scattering, thus directly changing the Earth's radiation budget (Boucher et al., 2013).





Estimates of direct radiative forcings by mineral dust are subject to large uncertainties, with global annual net (shortwave + longwave) radiative forcings at the surface having a cooling effect in the range of $(-0.5 \text{ to } -2.0)\,\mathrm{Wm}^{-2}$ (Choobari et al., 2014). Additionally, mineral dust particles can act as cloud condensation nuclei and ice nuclei, consequently influencing the formation of cloud droplets and ice crystals, resulting in additional climate modifications (e.g., Hendricks et al., 2011; Boucher

et al., 2013; Mülmenstädt and Feingold, 2018). These indirect effects of mineral dust on the Earth's radiation budget are even more uncertain than direct radiative forcings and are subject of ongoing research activities (Choobari et al., 2014; Tang et al., 2016; Mülmenstädt and Feingold, 2018). Dust storms also pose significant hazards for global air traffic (e.g., De Villiers and Heerden, 2007) and influence energy production of solar energy power plants (e.g., Rieger et al., 2017). Furthermore, dust particles may have negative implications for human health, e.g. by causing respiratory diseases (Chan et al., 2008; Sajani et al.,

2011; Giannadaki et al., 2014). On the other hand, mineral dust provides nutrients such as iron or phosphorus that are essential for the growth of tropical rainforests, as well as oceanic life (Chadwick et al., 1999; Jickells et al., 2005; Nenes et al., 2011; Yu et al., 2015).

To correctly simulate mineral dust in global models, a reliable representation of the particle numbers, the size distribution and the global distribution of dust particles is necessary. As mineral dust is a primary aerosol, dust abundance and distribution in the

atmosphere are strongly related to its emissions. In many GCCMs, mineral dust emissions are represented by climatologies, i.e. prescribed monthly mean dust emissions for a specific year (e.g., de Meij et al., 2006; Liu et al., 2007). The AeroCom project (Aerosol Comparison between Observations and Models) led to the development of a global dust emission climatology (Ginoux et al., 2001, 2004; Dentener et al., 2006), that has been widely used in global modelling studies (e.g. Huneeus et al., 2011). To simplify the description of dust emissions in global models, the climatology prescribes monthly mean emission

rates, neglecting the variation of emission fluxes on shorter time scales. However, dust emissions are strongly influenced by meteorology resulting in high temporal variability from day to day (e.g. dust storms). Dust emissions also show large long-term (e.g. year-to-year) variations (Mahowald et al., 2010; Banks et al., 2017). The AeroCom dust climatology, however, is representative of the year 2000, which was characterized by relatively low dust emissions (Weinzierl et al., 2017). It has been argued that using monthly mean dust climatologies in GCCMs could lead to a misrepresentation of strong dust outbreaks,

resulting in a negative bias of model dust concentrations during these episodes compared to observations (Aquila et al., 2011; Huneeus et al., 2011; Kaiser et al., 2019).

As an alternative to such offline dust emission climatologies, online parametrizations have been developed that account for temporal variability by calculating dust emissions from local surface wind velocities in each model timestep (e.g., Tegen et al., 2002; Balkanski et al., 2004). Several online dust emission schemes have been successfully implemented in GCCMs and have

been shown to adequately simulate global dust distribution patterns on daily, seasonal and multiannual timescales (Stier et al., 2005; Astitha et al., 2012; Gläser et al., 2012). However, online dust parametrizations also suffer from drawbacks. For example, they need to be tuned for every model setup according to a reference emission climatology by setting specific tuning parameters employed in the calculation of dust emission fluxes (e.g., Tegen et al., 2004). This is necessary to keep the total dust emissions comparable between different model simulations.





In this study, we aim to improve the representation of atmospheric mineral dust in the atmospheric chemistry general circulation model EMAC (ECHAM/MESSy Atmospheric Chemistry model; Jöckel et al., 2010, 2016) including the MESSy (Modular Earth Submodel System; Jöckel et al., 2010) aerosol microphysics submodel MADE3 (Modal Aerosol Dynamics model for Europe, adapted for global applications, 3rd generation; Kaiser et al., 2014). In previous model studies with MADE3 (or its

predecessors) in EMAC, dust emissions were represented by the offline AeroCom dust climatology (Aquila et al., 2011; Righi et al., 2013; Kaiser et al., 2019). We now apply the online dust emission scheme developed by Tegen et al. (2002) to account for highly variable wind-driven dust emissions and strong emission episodes. We compare results from simulations using the AeroCom dust climatology with those applying the online Tegen et al. (2002) emission scheme with respect to dust aerosol concentrations near source regions and in target regions of long range transport. Additionally, we analyse the effect of different

vertical and horizontal model resolutions, as well as the effect of varying the dust size distribution upon emission for the Tegen et al. (2002) dust setup. We analyse the capabilities of these different model setups with special focus on the representation of dust emissions as well as the resulting atmospheric dust distribution and properties. The objective is to improve the representation of mineral dust in the model and to optimize the model setup for future studies concerning, for instance, the effect of heterogeneous ice nucleation induced by ice nucleating particles such as mineral dust. As shown in many laboratory studies,

dust particles have indeed the ability to serve as very efficient ice nuclei (e.g. Kanji et al., 2017). The resulting potential of dust to influence ice clouds on the global scale has also been demonstrated by modelling studies (Lohmann and Diehl, 2006; Hoose et al., 2010; Hendricks et al., 2011). As future applications of our model are intended to focus on aerosol effects on ice cloud properties (Righi et al., 2020), the present study is a necessary step towards an improved model setup suitable for this kind of model investigations.

The model results obtained here are evaluated by comparison with different observations, i.e. ground-based remote sensing and airborne in situ measurements. In Kaiser et al. (2019) a thorough evaluation of different aerosol properties simulated with MADE3 as part of EMAC was performed. Here the model evaluation concentrates on measurements specifically related to mineral dust since it is the major target of the model improvements in this study. As a special focus, we compare the model results with data from the SALTRACE campaign, performed during June/July 2013 with observations in Barbados, Puerto Rico

and Cabo Verde (Weinzierl et al., 2017). SALTRACE aimed to explore the relevant processes associated with the transport of Saharan mineral dust across the Atlantic Ocean and its impacts on clouds and radiation. The Sahara Desert is the largest dust source on Earth providing at least half of the globally emitted dust (Huneeus et al., 2011). Data from the SALTRACE campaign is particularly extensive, including different measurement techniques and instruments. Foci were on dust source regions in the Sahara, dispersion and transformation processes, and long range dust transport towards the Caribbean, making the campaign

exceptionally valuable for our model evaluation. We simulate specific episodes of the SALTRACE campaign. For this episodic simulations, various meteorological model variables are nudged towards ECMWF reanalyses and transient aerosol emissions are prescribed for the corresponding time period. This enables us to directly compare our model results with the observations. In our previous studies (Aquila et al., 2011; Righi et al., 2013; Kaiser et al., 2019), a climatological simulation concept was applied instead of modelling a specific episode. There the comparison of long-term model means with short-term measurement

episodes led to discrepancies, due to different meteorological situations and emissions. The episodic comparison performed in





this study aims to reduce these uncertainties. In addition to the SALTRACE data, we apply long-term observations of aerosol optical depth from AERONET stations (Holben et al., 1998, 2001) at dust-dominated locations, covering also the SALTRACE episode, in order to evaluate the model's capability to reproduce the temporal variability of airborne mineral dust.

The paper is organized as follows. In Sect. 2 we describe the EMAC model, including the different model setups used in this work, as well as the observational data used for model evaluation. Results of the model evaluation are presented in Sect. 3. There, we first describe model results evaluated against AERONET station data, showing an improved representation of the temporal variability of mineral dust when applying the Tegen et al. (2002) dust parametrization. Secondly, we show that increasing the horizontal and vertical model resolution results in a better representation of the spatial distribution of mineral dust in the model when evaluated against SALTRACE campaign data. The main conclusions of this study are highlighted in Sect. 4.

## 2 Model description and observational data

### 2.1 EMAC setup

The EMAC model is a global numerical chemistry and climate simulation system including various submodels that describe tropospheric and middle atmosphere processes. It uses the second version of MESSy to connect multi-institutional computer codes. The core atmospheric model is the ECHAM5 (5th generation European Centre Hamburg) general circulation model (Roeckner et al., 2006).

In this work we apply EMAC (ECHAM5 version 5.3.02, MESSy version 2.54) in three different resolutions, namely T42L19, T42L31, and T63L31 with spherical truncations of T42 (corresponding to a quadratic Gaussian grid of approx. 2.8 by 2.8 degrees in latitude and longitude) and T63 (approx. 1.9 by 1.9 degrees), respectively, and with 19 or 31 vertical hybrid pressure levels up to $10\,\mathrm{hPa}$. Model timesteps for these resolutions are 30 minutes, 20 minutes, and 12 minutes respectively and the temporal resolution for most simulation output is chosen as 12 hours. The model output for aerosol optical depth (AOD) is generated every hour for comparisons with observations on a daily mean basis.

The EMAC-MADE3 setup used in this work is largely based on the setup described in Kaiser et al. (2019). In addition to the MESSy submodels used in their work, the diagnostic submodel S4D (Sampling in 4 Dimensions, Jöckel et al., 2010) is included here in order to extract model output along aircraft trajectories of the flights conducted during the SALTRACE campaign. The S4D submodel interpolates the model output along the track of a moving platform (here an aircraft) online, i.e. during the model simulation, thus facilitating a direct and more accurate comparison of model output and aircraft observations.

All simulations discussed in this paper cover the years 1999 to 2013 and were performed in nudged mode, i.e. wind divergence and vorticity, sea surface and land temperature, as well as the logarithm of the surface pressure were relaxed towards ECMWF reanalyses (ERA-Interim) for the corresponding years. The first simulated year (1999) is regarded as a spin-up phase and only the subsequent time period (2000–2013) is used for model evaluation. A summary and short description of the different simulation setups applied in this study is shown in Table 1.





**Table 1.** Summary of the different EMAC simulation setups applied in this study. All simulations cover the years 2000–2013.

| Abbreviation | Model resolution | Representation of dust emissions |
|---|---|---|
| T42L19Tegen | T42L19 | Tegen et al. (2002) online calculated dust |
| T42L31Tegen | T42L31 | Tegen et al. (2002) online calculated dust |
| T63L31Tegen | T63L31 | Tegen et al. (2002) online calculated dust |
| T42L31AeroCom | T42L31 | Prescribed year 2000 monthly mean dust emissions (AeroCom climatology) |
| T42L31TegenS | T42L31 | Tegen et al. (2002) online calculated dust, different size distribution of emitted dust [a] |

[a] The size distribution of mineral dust measured during the SAMUM-1 campaign was used (Weinzierl et al., 2009, 2011), see Sect. 2.3 for more details.

## 2.2 The aerosol submodel MADE3

MADE3 (Modal Aerosol Dynamics model for Europe, adapted for global applications, 3rd generation) was described in detail by Kaiser et al. (2014). Here we recall only its main aspects, as shown in Fig. 1. MADE3 simulates nine different aerosol species: sulfate ($SO_4$), ammonium ($NH_4$), nitrate ($NO_3$), sea-spray components other than chloride (mainly sodium; $Na$),

chloride ($Cl$), particulate organic matter (POM), black carbon (BC), mineral dust (DU), and aerosol water ($H_2O$).

These aerosol components are distributed into nine log-normal modes that represent different particle sizes and mixing states. Each of the MADE3 Aitken-, accumulation- and coarse-mode size ranges incorporates three modes for different particle mixing states: particles fully composed of water-soluble components, particles mainly composed of insoluble material (i.e. insoluble particles with only very thin coatings of soluble material), and mixed particles (i.e. soluble material with inclusions

of insoluble particles).

MADE3 simulates the following aerosol processes: gas-particle partitioning of semi-volatile species, particle coagulation, condensation of sulfuric acid and low-volatile secondary organic aerosol species, and new particle formation. MADE3 calculates changes of particle number concentration, size distribution, and particle composition induced by these processes and solves the aerosol dynamics equations by applying analytical approximations and process-specific numerical solvers. A de-

tailed description of this approach can be found in Kaiser et al. (2014).

## 2.3 Emission setup

The emission setup used in the present study is based in large parts on the setup of Kaiser et al. (2019), but in contrast to prescribed monthly anthropogenic and biomass burning emissions representative of the year 2000 (Lamarque et al., 2010), here we use transient prescribed emissions matching the simulated time period (2000-2013). This is important for direct comparability

of the model results with observations during the SALTRACE campaign. For the transient monthly anthropogenic emissions we use a combination of ACCMIP (Atmospheric Chemistry and Climate - Model Intercomparison Project, Lamarque et al.,





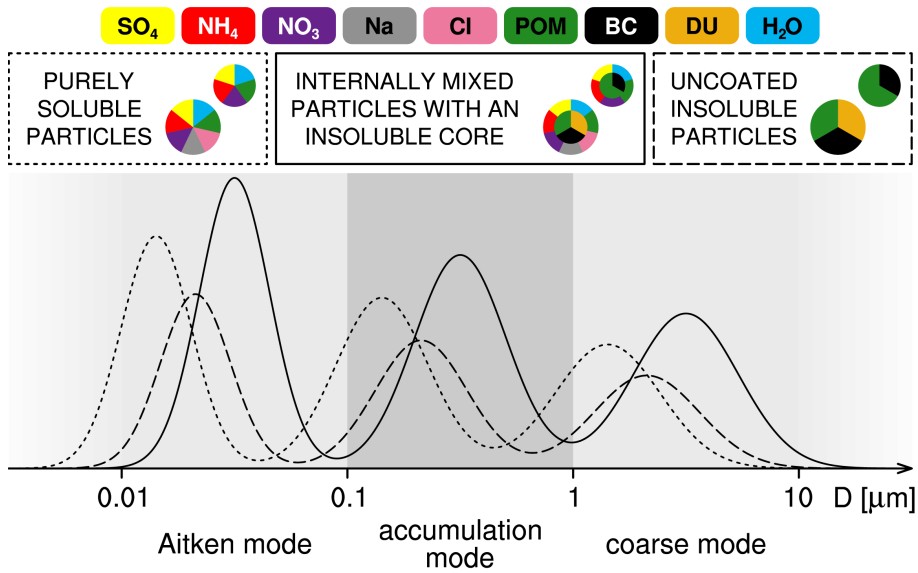

**Figure 1.** Schematic representation of the MADE3 submodel, as shown in Kaiser et al. (2019). The colors represent the different chemical components. The dotted, solid, and dashed lines correspond to the different mixing states (soluble, mixed, and insoluble, respectively)

2010) and RCP8.5 data (Representative Concentration Pathway leading to a radiative forcing of $8.5\,\mathrm{W/m^2}$, Riahi et al., 2007, 2011). Biomass burning emissions were taken from the Global Fire Emissions Database version 4 (GFED4s, van der Werf et al., 2017).

As described above, while Kaiser et al. (2019) used prescribed monthly mean dust emissions from the AeroCom offline

climatology, described in Dentener et al. (2006), we now apply the dust parametrization developed by Tegen et al. (2002), that calculates dust emissions online for every model timestep. Dust emissions are calculated for 192 internal dust size classes ranging from $0.2\,\mathrm{\mu m}$ to $1300\,\mathrm{\mu m}$ diameter according to the simulated $10\,\mathrm{m}$ wind velocity and prescribed external input fields of dust source areas, soil types and vegetation cover (for details see Tegen et al., 2002; Stier et al., 2005; Cheng et al., 2008; Gläser et al., 2012). The horizontal emission flux (HF) is calculated for each dust size class $i$ as:

$$\mathrm{HF}(i) = \frac{\rho_{\mathrm{air}}}{g} \cdot u^3 \cdot \left(1 + \frac{u_{\mathrm{thr}}(i)}{u}\right) \cdot \left(1 + \frac{u_{\mathrm{thr}}^2(i)}{u^2}\right) \cdot s_i \,, \ \ \text{if } u > u_{\mathrm{thr}}(i) \,, \ \ (\text{otherwise } \mathrm{HF}(i) = 0) \,, \tag{1}$$

with the density of air $\rho_{\mathrm{air}}$, the gravitational constant $g$, the relative surface area coverage for each size class $s_i$, the wind friction velocity $u$, which is calculated from the prognostic $10\,\mathrm{m}$ wind speed, and the threshold friction velocity $u_{\mathrm{thr}}(i)$. Only for velocities exceeding this threshold, dust emissions can occur. The vertical emission fluxes $\mathrm{VF}(i)$ are calculated from the horizontal fluxes according to:

$$\mathrm{VF}(i) = \alpha \cdot \beta \cdot f(\mathrm{LAI}) \cdot \mathrm{HF}(i) \,, \tag{2}$$

where $\alpha$ accounts for the soil texture characteristics, $\beta$ considers relative soil humidity and is 0 for soil humidities higher than 0.99 and 1 otherwise, and $f$ is a function of the Leaf Area Index (LAI) describing the vegetation cover.





To account for the log-normal representation of the aerosol size distribution in modal aerosol models like MADE3, the mass emission fluxes of the single size classes are summed up and distributed in two size modes that are here assigned to the MADE3 insoluble accumulation and coarse mode. As MADE3 also requires the corresponding number emissions, these are derived from mass emissions assuming a log-normal size distribution with count median diameter $D = 0.42\,\mu m$ and mode

width $\sigma = 1.59$ for the accumulation mode, and $D = 1.3\,\mu m$, $\sigma = 2.0$ for the coarse mode, respectively, following the AeroCom recommendations (Dentener et al., 2006). The corresponding conversion function (M2N) for log-normal distributions is given as (e.g. Seinfeld and Pandis, 2016):

$$\mathrm{M2N}_i(D_i, \sigma_i) = \frac{6}{\pi} \frac{1}{D_i^3 \exp(4.5 \ln^2 \sigma_i)\, \rho}\;, \tag{3}$$

with the median diameter $D_i$ and mode width $\sigma_i$ of the log-normal size distribution for mode $i$, and the density $\rho = 2500\,\mathrm{kg/m^3}$

of mineral dust.

In a sensitivity experiment (T42L31TegenS) we tested the effect of using a different assumption for the dust size distribution upon emission (results in Sect. 3.3), i.e. by varying the parameters for converting dust mass to number emissions. To this purpose, we use the dust size distribution measured during the SAMUM-1 dust campaign (Weinzierl et al., 2009, 2011). This campaign took place in 2006, in southern Morocco, close to the Sahara desert. It is therefore especially suited for this

sensitivity study, as it focuses on dust near the source regions in the Sahara. In Weinzierl et al. (2011) the dust size distribution is represented by four modes with $D_i$, $\sigma_i$, and the number concentration $N_i$, $i = 1, \ldots, 4$. The mass concentration $m_i$ of each of these four modes can be calculated using the factor $\mathrm{M2N}_i^{-1}$. For the online dust emission scheme a bimodal distribution is required. Therefore, the two smaller sized modes and the two larger ones are combined, in order to calculate the conversion factors for the accumulation ($\mathrm{M2N_{acc}}$) and the coarse mode ($\mathrm{M2N_{coa}}$) of the required bimodal distribution,

$$\mathrm{M2N_{acc}} = \frac{N_1 + N_2}{m_1 + m_2}, \quad \mathrm{M2N_{coa}} = \frac{N_3 + N_4}{m_3 + m_4}\;. \tag{4}$$

An overview of the mass-to-number conversion factors (M2N) for the different online dust model setups is shown in Table 2. Wind-driven online dust emissions need to be tuned for each applied model setup. The tuning procedure is described in the following section.

### 2.3.1 Dust emission tuning

In order to keep total wind-driven dust emissions comparable between different model simulations, dust emissions were tuned in the following way. As a reference for dust emissions we use the AeroCom climatology (Dentener et al., 2006), as this dataset is well evaluated and widely used in global modelling studies. We apply a global correction for online dust emissions by adjusting the wind friction velocity threshold for dust emissions by multiplication with the scaling factor $t_{\mathrm{wind}}$, as described in Tegen et al. (2004). Only for velocities exceeding this scaled threshold, dust emissions can occur. A higher (lower) threshold

therefore results in lower (higher) dust emissions. Emissions were tuned for the year 2000 in every model simulation, aiming to reproduce AeroCom emissions in the Saharan and Arabian desert region of $0°-40°$ N and $20°$ W $- 50°$ E, which amount to an annual dust emission of roughly $1200\,\mathrm{Tg}$. This region was selected because it is the largest dust source on the globe and



**Table 2.** Summary of wind stress threshold tuning parameter ($t_{\mathrm{wind}}$), orographic threshold for dust emission tuning ($t_{\mathrm{orogr}}$), mass-to-number conversion factors for accumulation and coarse mode ($M2N_{\mathrm{acc}}$, $M2N_{\mathrm{coa}}$), and resulting global and North Africa dust emissions of the year 2000, for the different online dust model setups.

| Model setup | $t_{\mathrm{wind}}$ | $t_{\mathrm{orogr}}$ (m) | $M2N_{\mathrm{acc}}$ (kg$^{-1}$) | $M2N_{\mathrm{coa}}$ (kg$^{-1}$) | Global emissions (Tg a$^{-1}$) | North Africa emissions [a] (Tg a$^{-1}$) |
|---|---|---|---|---|---|---|
| T42L19Tegen | 0.72 | 2500 | $3.92 \times 10^{15}$ | $4.0 \times 10^{13}$ | 2900 | 1210 |
| T42L31Tegen | 0.69 | 4000 | $3.92 \times 10^{15}$ | $4.0 \times 10^{13}$ | 1990 | 1230 |
| T63L31Tegen | 0.775 | 1770 | $3.92 \times 10^{15}$ | $4.0 \times 10^{13}$ | 1770 | 1270 |
| T42L31TegenS | 0.69 | 4000 | $5.79 \times 10^{16}$ | $1.16 \times 10^{13}$ | 2000 | 1240 |

[a] As the Tegen dust emissions were tuned to match AeroCom emissions over North Africa (1230 Tg a$^{-1}$), these values are almost identical.

because the SALTRACE dust campaign focuses on dust transport from North Africa to the Caribbean, which is a central point for model evaluation in this study. The resulting values for the wind stress threshold tuning parameter ($t_{\mathrm{wind}}$) are shown in Table 2.

Furthermore, an additional correction to dust emissions was necessary in our model, since it simulates unrealistically high
emissions in a few model grid boxes close to the Himalaya region. These artefacts dominate global dust emissions and are – e.g. for the T42L19 resolution – up to 100 times higher than emission peaks in the Sahara. In this critical region, dust sources, namely the Taklamakan desert, and areas of high surface winds (resulting from pronounced orographic gradients at the northern slope of the Himalayas) are located within the same model grid box. Hence, due to the relatively low spatial resolution, these areas overlap in the model, although they are spatially disjunct in reality. This conflict results in unrealistically high dust
emissions in the corresponding grid boxes and was also reported by Gläser et al. (2012) in a model study with EMAC using the Tegen et al. (2002) dust scheme. They further showed that these artefacts vanish for horizontal grid resolutions of and above T85 (approx. 1.4 by 1.4 degrees in latitude and longitude). As such a high resolution would be computationally too expensive and time consuming for our simulations and planned applications of this model setup, we choose a different solution.

In order to remove these high emission artefacts in the Himalaya region prior to the tuning procedure described above, we
exclude the corresponding grid boxes from the calculation of dust emissions by setting an upper threshold for orography. Above this threshold-height, emission fluxes are set to zero. The threshold value was adjusted for every model setup depending on the resolution, in order to target mostly the problematic grid boxes in the Himalaya region. Threshold values ($t_{\mathrm{orogr}}$) for the three different model resolutions are shown in Table 2. This procedure affects also some other grid boxes that show no high emission artefacts, mainly in the T42L19 and T63L31 setups, due to the somewhat lower $t_{\mathrm{orogr}}$ compared with T42L31. However, these
boxes are few and they correspond only to minor dust sources, mostly in the Tibetan Plateau. The numbers of dust emitting grid boxes that are excluded by setting $t_{\mathrm{orogr}}$ are 35, 12, 80, for the T42L19, T42L31, and T63L31 model setup, respectively. This procedure for tuning online dust emissions was also described and applied in Righi et al. (2020).





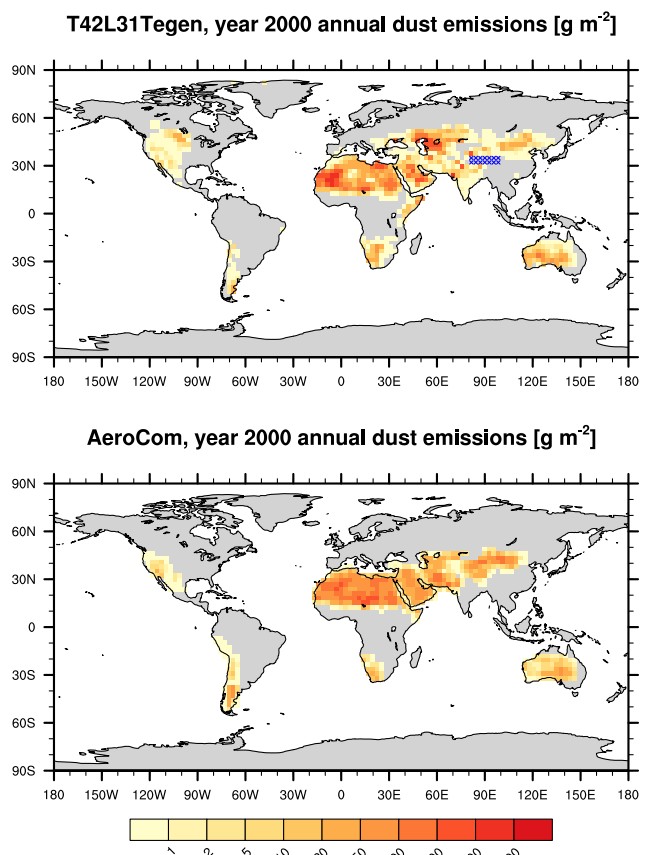

**Figure 2.** Global annual dust emissions in the T42L31Tegen (top) and T42L31AeroCom (bottom) setup. (Top) Emissions were corrected for artefacts in the Himalaya region and tuned according to AeroCom (for the region of $0° - 40°$ N and $20°$ W $- 50°$ E). Blue crosses correspond to excluded grid boxes due to setting $t_{\mathrm{orogr}}$ to $4000\,\mathrm{m}$ (12 of these 14 boxes would otherwise have emitted dust). The tuning results in a total dust emission of $1200\,\mathrm{Tg\,a^{-1}}$ in the Sahara region and a total global dust emission of $2000\,\mathrm{Tg\,a^{-1}}$ for the year 2000. (Bottom) Aerocom dust emissions were used for the T42L31AeroCom setup and as a reference for tuning online calculated Tegen et al. (2002) dust emissions. The total global AeroCom dust emission is $1700\,\mathrm{Tg\,a^{-1}}$.

The resulting tuned dust emissions of the year 2000 are shown in Fig. 2 for the T42L31Tegen setup. Total emissions over North Africa were tuned to match total emissions in the AeroCom climatology (about $1200\,\mathrm{Tg\,a^{-1}}$). Total global dust emissions of $2000\,\mathrm{Tg\,a^{-1}}$ are also comparable to the AeroCom value ($1700\,\mathrm{Tg\,a^{-1}}$) and lie in the range of other model studies, which simulate dust emissions between $514\,\mathrm{Tg\,a^{-1}}$ and $4313\,\mathrm{Tg\,a^{-1}}$ (Huneeus et al., 2011). A summary of tuned dust emis-
5   sions for all online dust model setups is shown in Table 2.





## 2.4 Observational data

Aircraft measurements provide valuable insights in the vertical distribution of aerosol particles by measurements of particle concentrations along the aircraft flight trajectory. Here, we use observational data from the SALTRACE campaign (Weinzierl et al., 2017). During this campaign (June, July 2013), aircraft measurements of various parameters, including size-resolved

particle number and black carbon mass concentrations, were performed mainly in the regions around Cabo Verde, Puerto Rico, and Barbados. From this data set we use the integral particle number concentrations in the size ranges 0.3 µm - 1.0 µm and 0.7 µm - 50 µm and the total black carbon mass mixing ratios for the model evaluation. The particle number concentrations in the size range from about 0.3 µm to 1.0 µm were measured by a Grimm model 1.129 optical particle counter (SkyOPC). The SkyOPC was operated onboard the Falcon research aircraft of the German Aerospace Center (DLR) behind an isokinetic

aerosol inlet with an upper particle cutoff diameter of about 2.5 µm near ground level, decreasing to about 1.5 µm at an altitude of 10 km. depending on altitude. Detailed specifications and performance analyses for this instrument can be found in Bundke et al. (2015) and in Walser et al. (2017). Detection of particles larger than the inlet cutoff was done using a wing-mounted aerosol size spectrometer CAS-DPOL (cloud and aerosol spectrometer probe with depolarization detection by Droplet Measurement Technologies Inc., Longmont, CO, U.S.A.; Baumgardner et al., 2001) with a nominal size detection range between

0.7 µm and 50 µm. The aircraft measurements are compared to model output extracted along the aircraft flighttracks by spatial and temporal interpolation, to ensure direct comparability between observation and model data. Additionally, we use ground-based Lidar observations also collected during the SALTRACE campaign. In particular dust extinction coefficients at 532 nm, measured with a stationary Lidar system located on Barbados, provide valuable information directly related to mineral dust (Groß et al., 2015, 2016).

In addition to SALTRACE observations, we use sun photometer measurements of aerosol optical depth (AOD) at 440 nm from the ground-based AErosol RObotic NETwork (AERONET; Holben et al., 1998, 2001). AOD provides an integral measure of radiation extinction by the vertical aerosol column. In the EMAC model, AOD is computed from simulated aerosol properties (in the submodel AEROPT) and compared with daily mean AOD values from AERONET radiometers (at 440 nm). To compare with the model data, we use a nearest-neighbour approach by selecting the model grid box covering the station coordinates.

The observational data used in this study are summarized in Table 3.

## 3 Model evaluation

### 3.1 Effects of dust emission scheme

In this section, we compare model results from simulations employing the different dust emission representations. Differences between the simulations with the model setup including prescribed offline dust emissions (AeroCom climatology) and the setup

using the Tegen et al. (2002) online dust parametrization are described (Simulations T42L31AeroCom and T42L31Tegen, respectively). In particular, we compare simulated AOD values with data from ground-based AERONET stations, in order to evaluate the capability of the different model versions to represent the temporal variability of airborne mineral dust.





**Table 3.** Summary of relevant details and references of the observational datasets used for the evaluation of model results simulated with EMAC-MADE3. Numbers in brackets in the column "time" indicate the number of flights for aircraft measurements and the number of observation days for SALTRACE Lidar measurements.

| Name | Location | Time | Parameter | Reference |
|---|---|---|---|---|
| SALTRACE aircraft (East) | Cabo Verde | June 2013 (5) | Particle number | Weinzierl et al. (2017) |
| | | | BC mass | Schwarz et al. (2017) |
| SALTRACE aircraft (West) | Eastern Caribbean | June/July 2013 (13) | Particle number | Weinzierl et al. (2017) |
| | | | BC mass | Schwarz et al. (2017) |
| SALTRACE Lidar | Barbados | June/July 2013 (24) | Dust extinction | Groß et al. (2015) |
| AERONET stations | 17 stations [a] | 2009–2013 [a] | AOD (440 nm) | Holben et al. (1998) |

[a] AERONET data from various dust-dominated stations located in a region of $5°$ N – $40°$ N and $20°$ W – $50°$ E covering the time period 2009–2013 was used. A detailed description of the selection criterion is given in Sect. 3.1

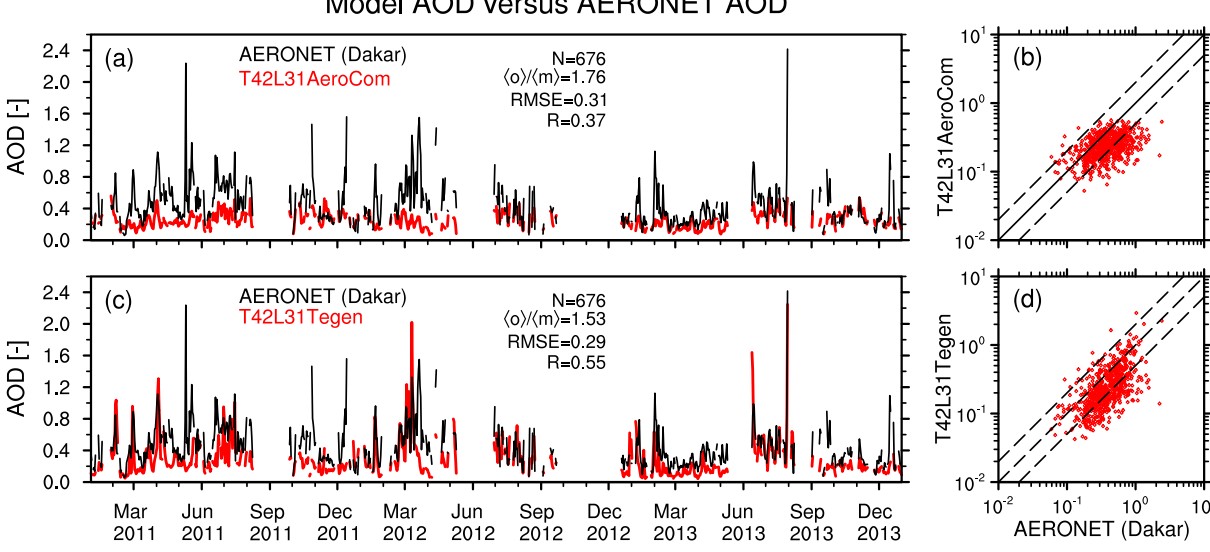

**Figure 3.** Model AOD versus AERONET station observations. Timeseries of AOD (at 440 nm) for the AERONET station located in Dakar (Senegal) are compared with model AOD on a daily mean basis for the time period 2011/01 – 2013/12. Subfigure (a) compares observation AOD (black line) with the offline dust model setup (T42L31AeroCom, red line). Gaps in the timeseries are due to missing observations on those days. Number of data points, ratio of averages of observation and model data, root mean square error (RMSE), and Pearson correlation coefficient (R) are shown. Subfigure (b) shows the data as scatterplot of model versus observation AOD data for the T42L31AeroCom model setup. Subfigure (c) and (d) are the same as (a) and (b) but show results from the online dust model setup (T42L31Tegen).

In Fig. 3, AOD timeseries of model results and observations are shown for the two model setups, i.e. with prescribed offline AeroCom dust emissions and online parametrized dust emissions, respectively. Apart from the representation of dust emissions,





the two model setups are identical. As an example, timeseries of daily averages for the AERONET station Dakar (Senegal) are shown for a period of 36 months (Jan 2011 – Dec 2013). Compared to the AeroCom setup, AOD peaks from observations are expectedly in most cases much better represented in the online dust setup, e.g. the correlation coefficient is increased from 0.37 to 0.55 and the root mean square error is reduced from 0.31 to 0.29. This can also be seen in Fig. 3b,d, where scatterplots

of model versus observation data for the two model setups are shown. Although total AOD is shown here (i.e. incorporating all types of aerosol particles), AOD peaks are probably related to strong dust events as the station is located in a dust-dominated region. This implies an improved representation of dust outbreaks when using the Tegen et al. (2002) online dust scheme.

For a statistical comparison, we compare simulated AOD with observations from all dust-dominated AERONET stations in a region of $5°$ N – $40°$ N and $20°$ W – $50°$ E, for the time period 2009 – 2013, on a daily average basis. We use the Ångstrom

exponent (AE, 870-440 nm) from AERONET measurements to select dust-dominated stations. An AE criterion is commonly used to extract the coarse-mode component from AOD data, which represents soil dust as the dominant coarse aerosol in desert regions (Ginoux et al., 2012; Eck et al., 1999; Parajuli et al., 2019). Stations with AE less than 0.75 (multi-annual mean) and with more than 50 observation days are selected. Their locations are shown in Fig. 4a.

To quantitatively compare model simulations with observational data, we use the skill score (S), defined by Taylor (2001):

$$S = \frac{4(1+R)^4}{(\frac{\sigma_{\mathrm{m}}}{\sigma_{\mathrm{o}}} + \frac{\sigma_{\mathrm{o}}}{\sigma_{\mathrm{m}}})^2(1+R_0)^4} \; , \tag{5}$$

where $R$ is the correlation coefficient, $\sigma_{\mathrm{m}}$ and $\sigma_{\mathrm{o}}$ are the standard deviations of model and observational data, respectively, and $R_0$ is the maximum attainable correlation. This skill score is commonly used for model comparisons with observations (e.g. Klingmüller et al., 2018; Parajuli et al., 2019). For simplicity, we use $R_0 = 1$, as we are mainly interested in the relative changes of the skill score for different model simulations. Skill score values range from 0 to 1, with higher values indicating a

better agreement between model and observations.

Fig. 4b shows the comparison of skill scores for the two model setups T42L31AeroCom and T42L31Tegen, respectively. In general nearly all selected AERONET stations show an improved agreement with model results for the Tegen et al. (2002) online dust setup compared to the offline dust setup. The average skill score over all stations is nearly twice as high for the T42L31Tegen setup (0.22) as for the T42L31Aerocom setup (0.14). Especially the Dakar station shows a nearly five times

higher skill score for the T42L31Tegen setup compared to T42L31AeroCom (0.38 versus 0.08, respectively). Remaining uncertainties and deviations from observed values can be attributed to spatial sampling issues when comparing grid-box averages to localized observations (Schutgens et al., 2016). Additional deviations may result from uncertainties in prescribed soil surface properties and modelled winds, as well as from assumptions on the specific optical properties of the single aerosol types in the AEROPT submodel, which are used to calculate AOD. Furthermore, the assumption on the dust size distribution upon

emission may lead to differences; this is analysed in Sect. 3.3 with a sensitivity experiment (T42L31TegenS).

### 3.2   Effects of model resolution

Previous EMAC studies employing the aerosol submodel MADE3 or its predecessors (Aquila et al., 2011; Righi et al., 2013, 2015, 2016; Kaiser et al., 2019) were mainly based on a relatively low model resolution of T42L19 (i.e. approx. 2.8 by 2.8



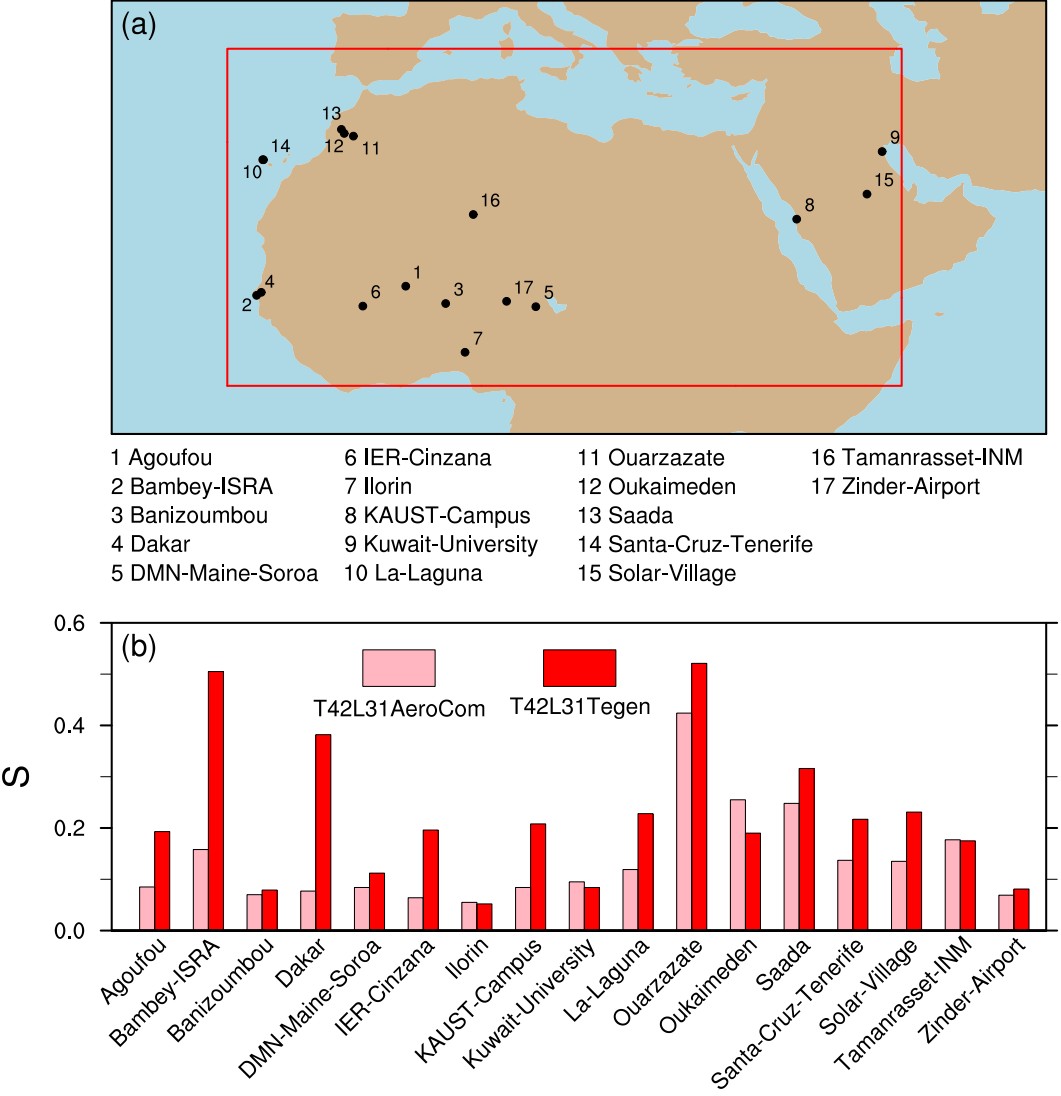

**Figure 4.** Locations of selected AERONET stations and skill scores for the T42L31AeroCom and T42L31Tegen setups. (a) AERONET stations in the region of $5°\,N-40°\,N$ and $20°\,W-50°\,E$ (red box), for the time period $2009-2013$, with Ångstrom exponents of $AE < 0.75$ (AE averaged over the time period $2009-2013$) and a minimum of 50 observation days were selected. (b) Skill scores (S) for these stations are calculated from AOD observations and model output for the T42L31AeroCom (pink bars) and T42L31Tegen setup (red bars), respectively.

degrees in latitude and longitude with 19 vertical levels up to $10\,\mathrm{hPa}$). In order to investigate the effect of the model resolution on dust emissions and transport with the Tegen et al. (2002) parametrization, we perform simulations with enhanced vertical (T42L31Tegen) and horizontal (T63L31Tegen) model resolution and compare them with the T42L19Tegen setup.





We compare the simulated vertical aerosol distribution with vertical aerosol concentration profiles measured during the SALTRACE campaign (Weinzierl et al., 2017). In general, comparing climatological 3-D model output with aircraft measurements is difficult and prone to large uncertainties due to the limited spatial and temporal data coverage of aircraft observations. In order to improve the climatological comparison method used in Kaiser et al. (2019), we constrained the model as described

in Sect. 2.1 to reproduce the large-scale meteorological conditions during the episode of the field campaign. We further employ the S4D submodel to extract model output along aircraft flight tracks online, i.e. during the model simulation, providing a more direct comparison of model output and aircraft observations, rather than by interpolating corresponding model values from the standard output. The aircraft observations have a time resolution of typically 1 to 10 seconds. For the evaluation, we vertically binned both the simulation and the measurement data into $1.6\,\mathrm{km}$ intervals. This enables a direct in situ-to-model comparison.

Additionally, we compare our model results with ground-based Lidar observations also collected during the SALTRACE campaign. In particular, we consider vertical profiles of dust extinction coefficients at $532\,\mathrm{nm}$, measured with a stationary Lidar system located on Barbados (Groß et al., 2015, 2016). Simulation and Lidar measurement data were binned into $500\,\mathrm{m}$ intervals for this comparison.

In Fig. 5 vertical aerosol profiles of total particle number concentrations in two different size ranges, as well as vertical pro-

files of the Lidar dust extinction coefficient are shown for the observations and the three different model setups, respectively. Only data from the SALTRACE-West regions (around Puerto Rico and Barbados) are presented here because of better data coverage due to a larger number of measurement flights compared to SALTRACE-East (around Cabo Verde). Number concentrations are shown for aerosol particles with diameters in the size range of $0.3\,\mathrm{\mu m} < \mathrm{D} < 1.0\,\mathrm{\mu m}$ and $0.7\,\mathrm{\mu m} < \mathrm{D} < 50\,\mathrm{\mu m}$, respectively. These size ranges represent the detection size limits of the particle counters used in the aircraft measurements and

serve as rough estimates for aerosol numbers in the accumulation and coarse mode, respectively. The size cutoff values of the particle counters are also subject to uncertainties and may change slightly during a flight.

In general, the low resolution T42L19 setup shows a reasonably good agreement with both aircraft and Lidar observations in the lower troposphere (up to around $600\,\mathrm{hPa}$) but overestimates number concentrations and extinction coefficients at higher altitudes significantly, up to a factor of 10 for the number concentration above $400\,\mathrm{hPa}$. This large positive bias is slightly

reduced for the T42L31 setup with higher vertical resolution. When increasing both the horizontal and the vertical model resolution (T63L31 setup) the bias at higher altitudes vanishes almost completely in the comparisons with number concentration measurements (Fig. 5a,b). Number concentrations are reduced by up to a factor of 10 compared to the T42L19 setup above $400\,\mathrm{hPa}$, so that they now correspond to observed values within the uncertainty ranges. Also, the steep gradient in the Lidar observations around $600\,\mathrm{hPa}$ (Fig. 5c) is reproduced better by the T63L31 setup, with again up to 10 times lower values

compared to the T42L19 setup. This steep decrease in the Lidar observations is representative of the vertical extent of the Saharan Air Layer (SAL), a warm, dry, elevated air layer (reaching up to approx. $4\,\mathrm{km}$ in the Caribbean) in which the main dust transport from the Sahara to the Caribbean takes place (Weinzierl et al., 2017; Haarig et al., 2019).

The comparison with Lidar observations is of special importance, as here the dust extinction coefficient provides a measure directly related to mineral dust, whereas in the total particle number concentrations also non-dust particles are included. Never-

theless, these size ranges comprising relatively large particles are probably dominated by mineral dust (Kaiser et al., 2019). The



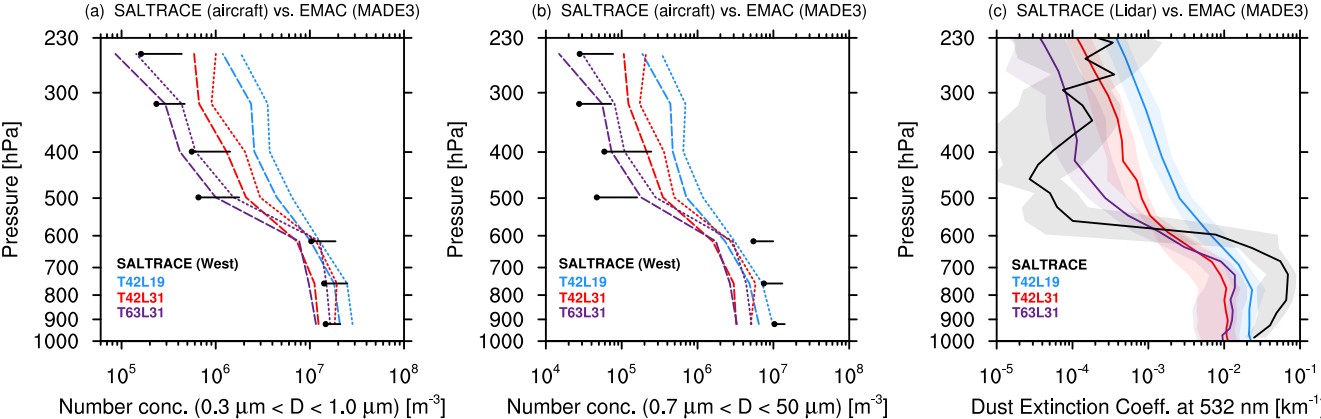

**Figure 5.** Comparisons between model results and different observational data from the SALTRACE mineral dust campaign (June, July 2013). (a) Data from aircraft measurements of total aerosol number concentration for particles with diameters in the range $0.3\,\mu m < D < 1.0\,\mu m$ is shown. Observational data from all SALTRACE-West flights (around Puerto Rico and Barbados) were binned into $1.6\,km$ height bins in order to create the vertical profiles. Dots represent mean values, whiskers represent standard deviations of observations (only positive direction shown). Mean values (long-dashed lines) and standard deviations (short-dashed lines) of the model results are shown for the three different resolutions: T42L19 (blue), T42L31 (red), T63L31 (purple). (b) Similar to (a), but for total aerosol number concentrations in the size range $0.7\,\mu m < D < 50\,\mu m$. (c) Simulated dust extinction coefficients compared with ground-based Lidar measurements at Barbados. Simulation and Lidar measurement data were binned into $500\,m$ intervals. Lines represent median values, shadings represent 25th-75th percentiles for the observations (black) and the three model setups (blue, red, and purple), respectively.

high bias of the T42L19 setup in the upper troposphere could be related to overestimated upward transport, possibly in convective plumes. This assumption is motivated by the fact that the convective top heights in the model (i.e. the uppermost model levels for convective transport) are on average approximately 15 percent higher in the T42L19 setup compared to T63L31 ($890\,hPa$ versus $780\,hPa$, also compared along the SALTRACE flight tracks). Another explanation for this strong positive bias could be an underestimation of aerosol scavenging through a too-low efficiency of the wet deposition processes in the model, as was also argued in Kaiser et al. (2019).

A similar evaluation of the vertical aerosol total particle number distribution as presented in Fig. 5a,b (SALTRACE-West region) was performed for SALTRACE-East (region around Cabo Verde; see Figure S1 in the Supplement). Those results show a similar behaviour as seen in Fig. 5a,b (SALTRACE-West), i.e. a large positive bias for the T42L19Tegen setup in the upper troposphere, which is reduced in the model configurations with higher spatial resolution (T42L31Tegen, T63L31Tegen). However, as only a few measurement flights were performed in that region, the data set is limited, which complicates the analysis and results in larger uncertainties.

In addition to measurements focusing on mineral dust, black carbon (BC) mass mixing ratios were measured during the SALTRACE campaign, likely representing aerosol particles originating from biomass burning events in Central Africa (Weinzierl et al., 2017). Hence, a similar comparison as for aerosol particle numbers can be performed for BC mass mixing

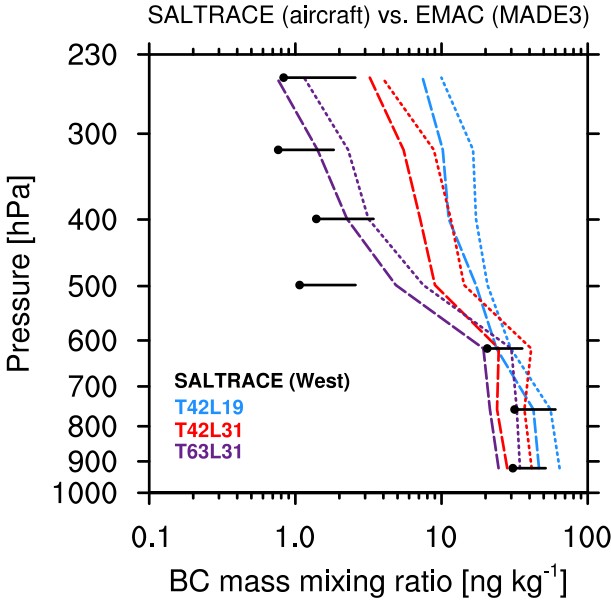

**Figure 6.** Black carbon mass mixing ratio profiles for SALTRACE observations and model results. Similar to the left and middle plot in Fig. 5, but for BC mass mixing ratios (in units of $\mathrm{ng\,kg^{-1}}$).

ratios (in units of $\mathrm{ng\,kg^{-1}}$) for the three different model setups (Fig. 6). Again, the high bias in the upper troposphere is significantly reduced for the T63L31Tegen setup with respect to T42L19Tegen, corroborating the findings described in the previous paragraphs.

Additionally, modelled BC mass mixing ratios, as well as number concentrations of particles in different size regimes were
evaluated against additional aircraft measurements from several campaigns, as done in Kaiser et al. (2019). The results are shown in Figures S3 and S4 in the Supplement. This evaluation is performed on a climatological basis, i.e. comparing long-term model monthly means with the observation campaign data, as described in detail by Kaiser et al. (2019). Results from the T63L31Tegen model setup (enhanced horizontal and vertical resolution with online calculated dust) are compared with the model results from Kaiser et al. (2019), i.e. T42L19 resolution with prescribed monthly mean AeroCom dust. For most
comparisons, the T63L31Tegen setup shows a better agreement with observations or only minimal changes compared with the Kaiser et al. (2019) simulation. This clearly shows that, beyond the representation of mineral dust, the enhanced model resolution generally improves the representation of the global aerosol.

### 3.3 Effects of size distribution assumptions

As described in Sect. 2.3, a typical mineral dust size distribution has to be assumed in the model in order to assign the
emitted dust particles to the respective lognormal size modes of the MADE3 aerosol submodel and also to convert mass emissions to number emissions. This assumption controls key properties of the freshly emitted particles, such as the dust particle number concentration in the specific modes or the ratio of fine to coarse mode dust particle number concentration.





Hence, it also has a large importance for modelling subsequent interactions of the particles with clouds and radiation. In order to analyse the sensitivity of the modelled atmospheric distribution and properties of mineral dust aerosols to an alternative size distribution assumption, we performed an additional sensitivity simulation (T42L31TegenS). In this experiment we apply the dust size distribution calculated from aircraft-based in situ measurements during the SAMUM campaign (Saharan Mineral

Dust Experiment) instead of the AeroCom size distribution (Dentener et al., 2006) used in the T42L31Tegen simulation. Within the SAMUM project, two field experiments were performed, which focused on the properties of airborne Sahara dust particles near the source regions (SAMUM-1, conducted in May/June 2006 in Morocco) and the properties of transported dust (SAMUM-2, conducted in January/February 2008 in the Cabo Verde area). For this sensitivity experiment we use the median dust size distribution from SAMUM-1 given in Weinzierl et al. (2011) which is based on numerous observations in elevated

dust layers over the source region between 19 May 2006 and 7 June 2006 (Weinzierl et al., 2009). There, the particle number size distribution of mineral dust aerosol measured during that field campaign is represented by a lognormal distribution with four modes. As a bimodal size distribution is required as input for the dust emission scheme in EMAC/MADE3, the two smaller sized modes of the measured distribution are combined, as well as the two modes with larger particles, to match the accumulation and coarse mode of MADE3, respectively.

We compare the simulation output from the T42L31Tegen and T42L31TegenS experiments with measurements from the SALTRACE campaign, similar to the evaluation in Sect. 3.2. Fig. 7 shows again aerosol number concentration profiles as well as vertical profiles of the Lidar extinction coefficient (as seen in Fig. 5), but comparing the T42L31Tegen and T42L31TegenS model setups. For the sensitivity simulation (T42L31TegenS), number concentrations of smaller sized particles are slightly shifted to larger values (Fig. 7a), whereas concentrations of larger particles are slightly decreased (Fig. 7b). This is in line with

the SAMUM-1 size distribution showing a larger (smaller) fraction of particles in the accumulation (coarse) mode, compared with the reference distribution (see also M2N values in Table 2). However, comparison of observed and simulated particle numbers is difficult, as the measured particle size ranges do not correspond directly to model accumulation and coarse mode. In the comparisons of dust extinction coefficients in Fig. 7c, the T42L31TegenS simulation shows smaller values. This is due to lower simulated dust mass concentrations compared with the reference simulation, resulting from stronger removal processes.

The lower coarse mode numbers of the SAMUM-1 distribution lead to larger simulated particle diameters, as the emitted dust mass remains constant. These larger particles are more efficiently removed by sedimentation and dry deposition processes in the model, with approximately 10 percent larger sedimentation and dry deposition fluxes in North Africa and the Caribbean. However, sedimentation of coarse particles is generally problematic for modal schemes, as size distributions may develop and deviate from the assumption of lognormal modes. Additionally, recent observations, in particular also during SALTRACE,

found coarse and giant particles large distances downwind of their sources (Weinzierl et al., 2017; Ryder et al., 2019). This could also hint to possibly missing processes in the model that keep large dust particles airborne over that long distances (Gasteiger et al., 2017).

In general, the differences between the two setups in Fig. 7 are small, with no notable improvement for the comparison with observations. Johnson et al. (2012) and Nabat et al. (2012) found improved agreement of simulated AOD with observations

when using a dust representation with a larger fraction of the dust mass emitted in the coarse mode. However, the SAMUM-





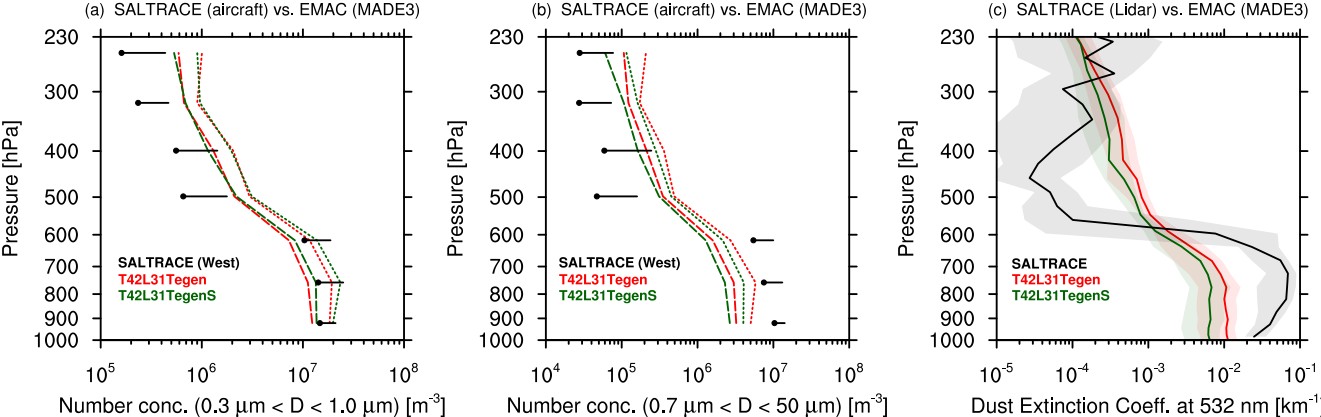

**Figure 7.** Similar to Fig. 5, but comparing the T42L31Tegen (red) and the T42L31TegenS (green) model setups with SALTRACE observations (black), i.e. the reference model setup and a setup with different assumptions for the size distribution of emitted dust. For the T42L31TegenS setup the dust size distribution calculated from measurements during the SAMUM-1 campaign is applied.

dust size distribution shows a larger fraction of emitted dust in the accumulation mode, compared with the reference size distribution. A comparison with AOD measurements from AERONET stations is shown in Fig. S2 in the Supplement and shows worse agreement for the T42L31TegenS simulation. Testing a size distribution with a larger fraction of dust particles in the coarse mode could be a subject for future studies.

## 4    Conclusions and outlook

In this paper, we use the aerosol microphysics submodel MADE3 as part of the atmospheric chemistry general circulation model EMAC and compare two different representations of mineral dust in the model. On the one hand, we use prescribed monthly dust emissions from the AeroCom climatology, as was also the case in the Kaiser et al. (2019) reference setup. On the other hand, we apply the Tegen et al. (2002) dust emission parametrization, where mineral dust emissions are calculated online for each model timestep. We compare the modelled aerosol optical depth at dust dominated locations with observations from the AERONET station network, and find that employing the Tegen et al. (2002) dust parametrization leads to improved agreement with observations compared with the offline dust model setup. Modelled AOD values show on average nearly twice as high skill scores when evaluated against several dust-dominated AERONET stations in North Africa (average skill score value of 0.22 for the online calculated dust setup versus 0.14 for the offline dust setup). This improvement is most likely due to a better representation of the highly variable wind-driven dust emissions and strong dust burst events.

Furthermore, we analyse the effect of increasing the horizontal and vertical model resolution on the dispersion of dust in the Tegen et al. (2002) dust emission model setup, by comparing the model results with ground-based Lidar remote sensing and aircraft measurements performed during the SALTRACE mineral dust campaign. Increasing the vertical (setup T42L31) and both the vertical and horizontal (setup T63L31) model resolution from a setup with a spherical truncation of T42 and 19





vertical hybrid pressure levels (setup T42L19) results in an improved agreement between model and observations, especially in the upper troposphere (above 400 hPa). The main improvement is achieved by increasing the horizontal model resolution from T42 to T63. Modelled particle number concentrations and dust extinction coefficients above 400 hPa decrease by up to a factor of 10, for the T63L31 setup versus T42L19. Overall, the long-range transport of mineral dust from North Africa

to the Caribbean, as well as the vertical transport into the upper troposphere is well represented in our model. Additionally, comparisons of modelled BC mass mixing ratios and particle number concentrations with aircraft measurements from several campaigns – as done in Kaiser et al. (2019) – show in most cases an improved model performance for the T63L31 setup compared to the results of Kaiser et al. (2019).

Finally, we tested the effect of varying the assumptions for the size distribution of emitted dust using the Tegen et al. (2002)

dust parametrization, by adopting the size distribution measured during the SAMUM-1 dust campaign (setup T42L31TegenS). However, we find no clear improvement with respect to the reference setup (T42L31-Tegen). Applying a size distribution with a larger fraction of dust particles in the coarse mode may improve the model results and could be a subject for future studies.

In general, we achieved an improved representation of atmospheric mineral dust in our model, especially due to an enhanced representation of dust emissions, compared with previous model setups. This provides an important foundation for future

model studies on the role of dust particles in the climate system including, for instance, simulations of the climatic impact of dust-induced modifications of mixed-phase and cirrus clouds.

*Code and data availability.* MESSy is continuously developed and applied by a consortium of institutions. The usage of MESSy, including MADE3, and access to the source code is licensed to all affiliates of institutions which are members of the MESSy Consortium. Institutions can become members of the MESSy Consortium by signing the MESSy Memorandum of Understanding. More information can be found

on the MESSy Consortium Website (http://www.messy-interface.org). The model configuration discussed in this paper has been developed based on version 2.54 and will be part of the next EMAC release (version 2.55).

The model simulation data analyzed in this work is available upon request and will be published via doi together with the final version of this manuscript.

*Author contributions.* CB conceived the study, implemented the method for tuning online dust emissions at low model resolutions, designed

and performed the simulations, analysed the data, evaluated and interpreted the results, and wrote the paper. JH contributed to conceiving the study, to the model evaluation, the interpretation of the results and to the text. MR assisted in preparing the simulation setup, helped designing the evaluation methods, and contributed to the interpretation of the results and to the text. BH and IT assisted in implementing the method for tuning online dust emissions at low model resolutions. DS, AW, and BW provided data from aircraft-based observations and assisted in the corresponding model evaluation. SG provided data from ground-based Lidar observations and assisted in the corresponding

model evaluation.





*Acknowledgements.* This study was supported by the DLR transport programme (projects *Global model studies on the effects of transport-induced aerosols on ice clouds and climate*, *Transport and the Environment - VEU2* and *Transport and Climate - TraK*), by the DLR space research programme (project *Climate relevant trace gases, aerosols and clouds - KliSAW*), by the German Federal Ministry for Economic

Affairs and Energy - BMWi (project *Digitally optimized Engineering for Services - DoEfS, contract no. 20X1701B*) and by the Initiative and Networking Fund of the Helmholtz Association (project *Advanced Earth System Modelling Capacity - ESM*). BW, AW, and DS would like to acknowledge funding from the Helmholtz Association under Grant VH-NG-606 (Helmholtz-Hochschul-Nachwuchsforschergruppe *AerCARE*). Furthermore, BW and AW have received funding from the European Research Council (ERC) under the European Union's Horizon 2020 research and innovation framework program/ERC Grant Agreement 640458 (*A-LIFE*). The EMAC simulations were preformed

at the German Climate Computing Center (DKRZ, Hamburg, Germany). The SALTRACE aircraft measurements in the Caribbean were funded by the Helmholtz Association and DLR. The SALTRACE flights in the Cabo Verde region were funded through the DLR-internal project *Volcanic Ash Impact on the Air Transport System (VolcATS)*. We are grateful to George Craig (LMU, Germany), Robert Sausen, Patrick Jöckel, Axel Lauer, Helmut Ziereis (DLR, Germany), and Klaus Klingmüller (MPI-C, Germany) for helpful discussions. The Earth System Model eValuation Tool (ESMValTool) v1.1.0 assisted in evaluating model results. We thank the AERONET PIs and Co-Is and their

staff for establishing and maintaining the 17 sites used in this investigation. We are grateful for the support of the whole MESSy team of developers and maintainers.





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
