# Peer review of "Modelling mineral dust emissions and atmospheric dispersion with MADE3 in EMAC v2.54"

_Geoscientific Model Development, 2020_

## Short Comment (SC1) · 23 Apr 2020

Dear authors,

in my role as Executive Editor of GMD I would like to bring to your attention, that the GMD rules require a permantent archiving of exactly that model code, with which the result published in an article have been produced. Thus if your model developments are based on MESSy version 2.54 and will be available in version 2.55 you still have to make the information available how the intermediate version the results were produced with, can be accessed.

Best regrads„

Astrid Kerkweg

---

## Referee Comment (RC1) · Anonymous Referee #1 · 9 Jun 2020

The manuscript present a series of sensitivity tests with the aerosol component in the EMAC model, by using observational data from the SALTRACE campaign and from the AERONET network as a benchmark. In particular the authors declare their main aim of evaluating the model performance when using prescribed, monthly aerosol fields, rather than an online dust emission scheme. Secondarily, they also evaluate the effects of model resolution and prescription of dust size distribution at emission. The scope of this study is relevant and the work is generally well organized and well presented. In my opinion there's a couple of issues that need to be addressed, and a few aspects to be clarified.

I would expect the AeroCom dust climatology and the monthly average dust fields generated by EMAC with Tegen dust emission scheme to have some (possibly significant)

[Figure]

differences. Therefore the comparison might reflect those differences as well, and in fact it might as well tell us more about a combination of information e.g. the 2000 climatology bearing some resemblance to the average climatology or with the meteorological conditions during SALTRACE, or that the Tegen-EMAC model yielding better results than the AeroCom average, etc. A more direct comparison to assess the effect of time averaging would be to use the monthly average dust fields generated by EMAC with Tegen dust emission scheme, as an offline prescribed dust field instead of the AeroCom one.

I would recommend a more "varied" pool of references, and in general an introduction and discussion that relate more extensively to the existing literature.

Can you comment on the optical properties you use? It would be relevant here to show something about your mass extinction efficiency at least.

(p. 4, 21-22) Do you write time-integrated or instantaneous variables as output?

(p.7, 5-) It would be good to provide the mass median diameter (or show a plot) for the two modes at emissions in the main case and in the sensitivity study, so that we can make sense of the information more clearly in relation to relevant existing literature.

---

## Referee Comment (RC2) · Anonymous Referee #2 · 9 Jun 2020

The article presents EMAC model results for atmospheric mineral dust which are evaluated using aircraft, Lidar and sun photometer observations. The authors combine the MADE3 aerosol microphysics submodel with the online dust emission scheme by Tegen et. al.. While both have been presented and implemented in previous studies, to my knowledge the combined application is new.

The evaluation makes use of measurements from the SALTRACE campaign which focussed on Saharan dust, and AERONET retrievals from stations in and surrounding the Sahara. Consequently the Sahara as the globally most important dust source is well represented, but other sources, e.g., the Asian deserts, are excluded from the evaluation. In this regard I would have appreciated a more complete evaluation, considering that EMAC is a global model and most often used for global studies. Nevertheless, I

[Printer-friendly version](javascript:void(0))

[Discussion paper](javascript:void(0))

[Figure]

believe that in the present form the article serves the authors' purpose of guiding future model setups.

Therefore I recommend to publish the article, which is generally well organised and written, after addressing the following comments.

Section 1

Given the modular nature of EMAC, to put this study into the context of existing EMAC studies, it might be worth to briefly relate MADE3 to other aerosol submodels such as GMXe.

Section 2.2

What are the size ranges used for the Aitken-, accumulation- and coarse modes?

Figure 1

Specifying which line corresponds to which mixing state suggests some meaning in the relative locations of the maxima, but supposedly the distributions are just examples? Do the grey shades indicate some thresholds at 0.1 and 1 um?

Equation (1)

"$(1 + u\_thr^2(i) / u^2)$" should read "$(1 - u\_thr^2(i) / u^2)$"

Page 6, line 16

Please define how you quantify soil humidity, in particular please mention the meaning of the value 0.99, is it the fraction of the field capacity?

Page 7, line 2

Please be more specific, I assume that not all size classes are summed to a single flux and distributed into the two modes only afterwards, but that some classes are summed to obtain the accumulation mode flux, another sum yields the coarse mode flux.

Page 7, line 5

Better use "sigma_g" instead of "sigma" and introduce as "[...] geometric standard deviation sigma_g = 1.59 [...]".

Figure 3

The time period used for this plot is shorter than the AERONET period specified in Table 3. If this is just for clarity, it should be pointed out in the caption to avoid the suspicion of cherry picking. For the same reason it should be mentioned that the plot shows data from the station which benefits most from the online dust emissions. I suggest to include the corresponding plots for the other stations of Figure 4, using the full 5 years, in the supplement.

Section 3.3

A direct comparison of the two emission size distributions (reference and SAMUM-1) would be helpful, particularly because it is not immediately clear what diameters Eq. (4) produces.

Figure 7

Since it is taken from aircraft measurements, do you expect the SAMUM-1 size distribution to be already affected by transport from emissions to observations? This would mean a slight bias towards smaller particles, consistent with the figure.

---

## Referee Comment (RC3) · Anonymous Referee #3 · 18 Jun 2020

This manuscript, submitted to Geoscientific Model Development, presents a sensitivity study on the representation of dust emissions in the atmospheric chemistry general circulation model EMAC. The authors focus (1) on the use of dynamic dust emission instead of climatological emissions, (2) the impact of increasing vertical and horizontal resolution, and (3) the size distribution of the emitted dust.

Results show a general improvement in the representation of dust aerosols with the use of dynamic dust emissions instead of climatologies. The increased resolution is also helpful in reproducing the vertical distribution of dust aerosols, while the changes on the size distributions have less impact. All these results are interesting and provide important conclusions for model developers. The paper is generally well organized and written, but the following comments need to be taken into account before considering

a publication in GMD.

Main comments:

1) The title is too general in relation to what is really dealt with in the text. I suggest to mention explicitly the word "emissions", and possibly also a reference to the sensitivity tests (dynamic emissions, impact of resolution, size distribution).

2) I found that the abstract needs to be rewritten to put forward the main conclusions of the paper, and give more information on the tests realized in this study. For example, the resolutions tested here should be explicitly mentioned, and more information about the results on the size distribution could be added. Besides, the first two sentences look more like an introduction than a summary, and could therefore be deleted.

3) The different resolutions analyzed in this study (between 19 and 31 vertical levels, and between 1.9 and 2.8 degrees) are quite coarse compared to other climate simulations. Could you comment on this point, and argue if these results could be relevant for finer resolutions?

4) This paper is focusing on dust aerosols, I do not understand why you present a comparison with black carbon (BC) concentrations in Figure 6. This could confuse the purpose of the paper. I suggest to remove Figure 6 and description on BC, or at least moving it to supplementary material.

Specific comments :

1) Page 2 line 15: "in many GCCMs, mineral dust emissions are represented by climatologies". I am not sure this is still true today, in particular in the recent CMIP6 simulations. Could you justify this statement with references?

2) Page 2 lines 31-34: The difficulty in assessing properly dust emissions (and not dust load) could be mentioned.

3) Page 4 line 30: Please add a reference for ERA-Interim.

4) Page 8 Table 2: Please clarify if the total emissions are given before or after the tuning. Does "North Africa" includes the Arabian desert region as mentioned in line 31 page 7?

5) Page 10 line 24: Have you tried to use another method than nearest-neighbour approach? Maybe you could interpolate the model grid on the location of the AERONET stations, which could avoid potential discontinuities between model grid boxes.

6) Page 11 Figure 3: Why have you used a log-scale for scatter plots?

7) Page 12 lines 21-25 and Figure 4: 3 stations (Ilorin, Kuwait-University and Oukaimeden) have lower skill scores with the Tegen emissions. Could you comment on this point?

8) Section 3.2: I wonder if the resolution also improves the AOD. It would be interesting to have the skill scores on AOD for the different simulations testing the horizontal and vertical resolutions, similar as what has been done in Figure 4 for the comparison between Aerocom and Tegen emissions.

Other corrections:

1) Page 2 lines 1-2: radiative forcing (without s)

2) Page 7: there is a section 2.3.1 without 2.3.2, could you check the numbering of subsections ?

3) Figure 5 page 15: Long-dashed and short-dashed lines are difficult to distinguish. Could you improve it?

―――――――――――――――――――――

---

## Author Comment (AC1) · 17 Jul 2020

Thank you for spotting this. We added the following text to the code availability section:

"The exact code version used to produce the results of this paper is archived at the German Climate Computing Center (DKRZ) and can be made available to members of the MESSy community upon request."

———————————————

---

## Author Comment (AC2) · 17 Jul 2020

**Modelling mineral dust emissions and atmospheric dispersion with MADE3 in EMAC v2.54**
**C. Beer et al.**
**Replies to referee comments**

We are grateful to the three reviewers for their important comments and constructive criticism. This greatly helped us improving the manuscript.

The comments of the single reviewers are addressed below, with the reviewer comments marked in blue, author replies in black, and text quotes in red.

**Anonymous referee 1**

1) I would expect the AeroCom dust climatology and the monthly average dust fields generated by EMAC with Tegen dust emission scheme to have some (possibly significant) differences. Therefore the comparison might reflect those differences as well, and in fact it might as well tell us more about a combination of information e.g. the 2000 climatology bearing some resemblance to the average climatology or with the meteorological conditions during SALTRACE, or that the Tegen-EMAC model yielding better results than the AeroCom average, etc. A more direct comparison to assess the effect of time averaging would be to use the monthly average dust fields generated by EMAC with Tegen dust emission scheme, as an offline prescribed dust field instead of the AeroCom one.

We thank the reviewer for this comment. The comparison of the new online dust emission setup (Tegen et al. 2002 scheme) with the previously used model setup applying the offline dust emission climatology (AeroCom) is the main focus of this study. However, as mentioned by the reviewer, a more detailed comparison of the online and offline dust emissions, including the seasonal cycle, could be helpful. Therefore, we included a figure with seasonal monthly mean dust emissions of the two model setups in the supplement, and included the following text in the revised version of the manuscript:

(page 9, line 4) […] The seasonal online dust emissions compare also reasonably well with the AeroCom climatology. However, the online emissions are strongest in the spring and summer months, while the AeroCom climatology shows the maximum in the winter season (see Fig. S1 in the Supplement). This deviation may be a result of the calculation of the wind-driven dust emissions, but could also be due to a possible atypical seasonal cycle for the year 2000.

2) I would recommend a more "varied" pool of references, and in general an introduction and discussion that relate more extensively to the existing literature.

We added several additional references to the text:

- Shao et al., 2011: page 2, line 14, "Dust cycle: An emerging core theme in Earth system science"
- Prakash et al., 2015: page 2, line 22, "The impact of dust storms on the Arabian Peninsula and the Red Sea"
- Hoose and Möhler, 2012: page 3, line15, "Heterogeneous ice nucleation on atmospheric aerosols: a review of results from laboratory experiments"
- Dee et al., 2011: page 4, line 30, "The ERA-Interim reanalysis: configuration and performance of the data assimilation system"
- Pringle et al., 2010: page 5, line 17, "Description and evaluation of GMXe: a new aerosol submodel for global simulations (v1)"
- Hess et al., 1998: page 10, line 1, "Optical Properties of Aerosols and Clouds: The Software Package OPAC"
- Koepke et al., 1997: page 10, line 1, "Global Aerosol Data Set"

3) Can you comment on the optical properties you use? It would be relevant here to show something about your mass extinction efficiency at least.

The aerosol optical properties are calculated in the EMAC submodel AEROPT according to input from OPAC (optical properties for aerosols and clouds, Hess et al., 1998). The OPAC package uses basic optical properties from Koepke et al., 1997. Optical properties for mineral dust are assumed according to Sahara conditions. We added the following text to the main paper:

(page 10, line 23-25) In the EMAC model, AOD is computed from simulated aerosol properties in the submodel AEROPT. AEROPT considers aerosol optical properties calculated according to the OPAC (optical properties for aerosols and clouds, Hess et al., 1998) software package, which follows the basic optical properties from Koepke et al. (1997).

4) (p. 4, 21-22) Do you write time-integrated or instantaneous variables as output?

For the analysis performed in this study we use time-averaged model output for AOD and instantaneous output otherwise, with a time frequency of 12 hours.

(page4, line 26) We use time-averaged model output for AOD and instantaneous output otherwise.

5) (p.7, 5-) It would be good to provide the mass median diameter (or show a plot) for the two modes at emissions in the main case and in the sensitivity study, so that we can make sense of the information more clearly in relation to relevant existing literature.

Thank you for this suggestion. We added a plot comparing the two size distributions to the supplement and added the following statement to the revised version:

(page 7, line 21) […] Additionally, we show the two number size distributions of the reference and the sensitivity study in Fig. S2 in the Supplement.

**Anonymous referee 2**

The evaluation makes use of measurements from the SALTRACE campaign which focussed on Saharan dust, and AERONET retrievals from stations in and surrounding the Sahara. Consequently the Sahara as the globally most important dust source is well represented, but other sources, e.g., the Asian deserts, are excluded from the evaluation. In this regard I would have appreciated a more complete evaluation, considering that EMAC is a global model and most often used for global studies. Nevertheless, I believe that in the present form the article serves the authors' purpose of guiding future model setups.

As stated by the reviewer, the representation of the North African dust emission regions is a main focus of this study, as they represent the most important global dust sources. Additionally, we performed a comparison of model and AERONET AOD data for stations in other regions (similar to Fig. 4) and included it in the Supplement. This analysis also shows an improvement for the online dust emission setup for most of these stations. We added the following text to the main paper:

(page 12, line 25) Additionally, a comparison with stations in other regions on the globe also shows improvements for most of these stations when using the online dust emission setup (see Fig. S4 in the Supplement).

1) Given the modular nature of EMAC, to put this study into the context of existing EMAC studies, it might be worth to briefly relate MADE3 to other aerosol submodels such as GMXe.

We thank the reviewer for the suggestion. The aerosol submodel GMXe (Pringle et al. 2010), is comparable to MADE3 but uses different numerical approaches. We added the following paragraph to the main text:

(page 5, line 15) A similar modal aerosol (MESSy-)submodel which is comparable to MADE3 is GMXe (Global Modal-aerosol eXtension, Pringle et al., 2010). A major difference between the two aerosol models is that MADE3 distinguishes between purely soluble particles and particles containing insoluble material, with the intention to enable a more straightforward quantification of the number concentrations of ice nucleating particles (Righi et al., 2020).

2) Section 2.2: What are the size ranges used for the Aitken-, accumulation- and coarse modes?
Typical size ranges of the MADE3 modes can be seen in Fig. 1. MADE3 uses not fixed but dynamical mode sizes, which may change during a simulation and are dependent on the assumption of the emitted and nucleated particle sizes.

3) Figure 1: Specifying which line corresponds to which mixing state suggests some meaning in the relative locations of the maxima, but supposedly the distributions are just examples? Do the grey shades indicate some thresholds at 0.1 and 1 um?
The distributions represent typical examples for the modes in MADE3. Grey shadings are to visually separate typical Aitken, accumulation and coarse mode sizes, but do not represent any specific thresholds in the model, which are set dynamically. We added this to the figure caption:
Grey shadings are to visually separate typical Aitken, accumulation and coarse mode sizes.

4) Equation 1: "(1 + u_thr^2(i) / u^2)" should read "(1 - u_thr^2(i) / u^2)"
Thank you for spotting this. It has been corrected in the revised version.

5) Page 6, line 16: Please define how you quantify soil humidity, in particular please mention the meaning of the value 0.99, is it the fraction of the field capacity?
Beta is zero if the upper layer soil moisture is at field capacity and 0.99 otherwise. Soil moisture is taken from ECHAM. We changed the variable name beta to I_theta and the text accordingly:
(page 6, line 16) I_theta is zero if the upper layer soil moisture is at field capacity and 0.99 otherwise.

6) Page 7, line 2: Please be more specific, I assume that not all size classes are summed to a single flux and distributed into the two modes only afterwards, but that some classes are summed to obtain the accumulation mode flux, another sum yields the coarse mode flux.
Thank you for this comment. We changed this in the text:
To account for the log-normal representation of the aerosol size distribution in modal aerosol models like MADE3, the internal emission fluxes for accumulation and coarse classes of the Tegen et al. 2002 parametrization are summed, respectively, and assigned to the MADE3 insoluble accumulation and coarse modes.

7) Page 7, line 5: Better use "sigma_g" instead of "sigma" and introduce as "[...] geometric standard deviation sigma_g = 1.59 [...]".
Thank you, we changed this as suggested.

8) Figure 3: The time period used for this plot is shorter than the AERONET period specified in Table 3. If this is just for clarity, it should be pointed out in the caption to avoid the suspicion of cherry picking. For the same reason it should be mentioned that the plot shows data from the station which benefits most from the online dust emissions. I suggest to include the corresponding plots for the other stations of Figure 4, using the full 5 years, in the supplement.
We thank the reviewer for mentioning this, and included a corresponding description to the figure caption and to the text:
(page 11) […] compared with model AOD on a daily mean basis for the time period 2011/01 – 2013/12. For clarity only a part of the full time period (starting in 2009/01) is shown here. […]
(page 12, line1) […] are shown for a period of 36 months (Jan 2011 – Dec 2013). The model results obtained for this station benefit most from applying the online dust emission scheme.
An additional figure showing time series of the other stations was added to the supplement and the following text was included in the main paper:
(page 12, line 7) In addition, AOD time series of other AERONET stations in North Africa and the Arabian Peninsula (see station locations in Fig. 4a) is shown in Fig. S3 in the Supplement. There, an improved representation of AOD peaks in the T42L31Tegen model setup is also visible for these additional stations.

9) Section 3.3: A direct comparison of the two emission size distributions (reference and SAMUM-1) would be helpful, particularly because it is not immediately clear what diameters Eq. (4) produces.

Thank you for this suggestion. We added a plot comparing the two size distributions to the supplement and added the following statement to the revised version:

(page 7, line 21) […] Additionally, we show the two number size distributions of the reference and the sensitivity study in Fig. S2 in the Supplement.

10) Figure 7: Since it is taken from aircraft measurements, do you expect the SAMUM-1 size distribution to be already affected by transport from emissions to observations? This would mean a slight bias towards smaller particles, consistent with the figure.

We thank the reviewer for this interesting suggestion. Indeed, as the SAMUM-1 campaign focused specifically on the dust source regions, where those flights took place, an effect of dust transport is probably small, but cannot be ruled out. We have added a corresponding statement to the text:

(page 18, line 4) Additionally, a slight bias towards smaller particles in the SAMUM-1 data could be due to effects of dust transport from emission to observation regions. However, as the flights took place near the source regions, this effect is probably small.

**Anonymous referee 3**

1) The title is too general in relation to what is really dealt with in the text. I suggest to mention explicitly the word "emissions", and possibly also a reference to the sensitivity tests (dynamic emissions, impact of resolution, size distribution).

Thank you for this comment. Besides mineral dust emissions, we also consider dust transport and atmospheric dispersion. A reference to sensitivity tests in the title would probably be too specific. We changed the title to: "Modelling mineral dust emissions and atmospheric dispersion with MADE3 in EMAC (v2.54)"

2) I found that the abstract needs to be rewritten to put forward the main conclusions of the paper, and give more information on the tests realized in this study. For example, the resolutions tested here should be explicitly mentioned, and more information about the results on the size distribution could be added. Besides, the first two sentences look more like an introduction than a summary, and could therefore be deleted.

Thank you for this suggestion. We changed the abstract accordingly.

It was hypothesized that using mineral dust emission climatologies in Global Chemistry Climate Models (GCCMs), i.e. prescribed monthly mean dust emissions representative of a specific year, may lead to misrepresentations of strong dust burst events. This could result in a negative bias of model dust concentrations compared to observations for these episodes. […] Furthermore, we analyse the effect of increasing the vertical and horizontal model resolution on mineral dust properties in our model. We compare results from simulations with T42L31 and T63L31 model resolution (2.8 by 2.8 degrees and 1.9 by 1.9 degrees in latitude and longitude, respectively, 31 vertical levels) with the reference setup (T42L19). […] Additionally, we analyse the effect of varying assumptions for the size distribution of emitted dust, but find only a weak sensitivity concerning these changes. […]

3) The different resolutions analyzed in this study (between 19 and 31 vertical levels, and between 1.9 and 2.8 degrees) are quite coarse compared to other climate simulations. Could you comment on this point, and argue if these results could be relevant for finer resolutions?

Even higher model resolutions could improve the representation of dust emissions further, as shown by e.g. Gläser et al., 2012. But as we aimed to find a model setup best suited for future applications e.g. multi-year climate simulations including ice nucleation on dust particles, higher model resolutions would not be feasible, due to the strongly increased computation costs and the need of performing many sensitivity simulations in process-oriented studies.

4) This paper is focusing on dust aerosols, I do not understand why you present a comparison with black carbon (BC) concentrations in Figure 6. This could confuse the purpose of the paper. I suggest to remove Figure 6 and description on BC, or at least moving it to supplementary material.

We find, that the improved representation of BC, besides DU, is an interesting aspect and important for future studies. But as this is not the main topic of this work we moved the respective figure to the supplement.

5) Page 2 line 15: "in many GCCMs, mineral dust emissions are represented by climatologies". I am not sure this is still true today, in particular in the recent CMIP6 simulations. Could you justify this statement with references?

Thank you for this comment. Indeed, most models today use online dust emission parametrizations (CMIP5, CMIP6). We changed the text accordingly. "A simple and straightforward way of representing dust emissions in GCCMs is the use of climatologies, i.e. prescribed monthly mean dust emissions for a specific year (e.g., de Meij et al., 2006; Liu et al., 2007)."

6) Page 2 lines 31-34: The difficulty in assessing properly dust emissions (and not dust load) could be mentioned

We agree and stated this as: "[…] Also, in contrast to observables like dust load, dust emissions are generally difficult to assess."

7) Page 4 line 30: Please add a reference for ERA-Interim.
We added this reference (Dee et al., 2011).

8) Page 8 Table 2: Please clarify if the total emissions are given before or after the tuning. Does "North Africa" include the Arabian desert region as mentioned in line 31 page 7?

The emissions given in Table 2 represent the values after the tuning procedure. Therefore the North Africa total emissions are the same for every model setup. They also include emissions from the Arabian Peninsula. We changed table and caption accordingly.

9) Page 10 line 24: Have you tried to use another method than nearest-neighbor approach? Maybe you could interpolate the model grid on the location of the AERONET stations, which could avoid potential discontinuities between model grid boxes.

Nearest-neighbor is the most "conservative" approach, as it refers to the values actually simulated by the model. Using an interpolation method assumes a uniform, linear gradient between adjacent grid boxes, which for 300 km grid box dimensions is not necessarily true.
Another advantage of the nearest-neighbor approach is an easier interpretation of the results, as you know which area your grid box is covering and that the value you are plotting is a mean over that area. We do not see the issue of the discontinuities, since we are comparing point locations along time.

10) Page 11 Figure 3: Why have you used a log-scale for scatter plots?
We find a logarithmic scale improves the visualization of the model-versus-observation comparison, also because the data cover a range of more than 1 order of magnitude.

11) Page 12 lines 21-25 and Figure 4: 3 stations (Ilorin, Kuwait-University and Oukaimeden) have lower skill scores with the Tegen emissions. Could you comment on this point?

The stations Ilorin and Kuwait-University show only slightly reduced skill scores, while many other stations have greatly improved in the Tegen setup. Hence, specific local conditions could be responsible for this inconsistency, rather than systematic model errors. The Oukaimeden station lies in close proximity to other stations that show improved skill scores, also it is located on a mountain (2760m elevation), which could explain the differences.

12) Section 3.2: I wonder if the resolution also improves the AOD. It would be interesting to have the skill scores on AOD for the different simulations testing the horizontal and vertical resolutions, similar as what has been done in Figure 4 for the comparison between AeroCom and Tegen emissions.

Thank you for this suggestion. We added an extra figure to the supplement, showing the AOD comparisons for the different model resolutions and added a corresponding discussion to the main paper.

(page 15, line 12) Additionally, we analyse the effect of increased model resolution on the AOD comparisons (as seen in Fig. 4). However, no clear improvement of the model comparison with AERONET AOD data is visible from this analysis (see Fig. S6 in the Supplement) , as the increase of model resolution probably mainly influences the representation of long-range transport and dust properties larger distances away from the source regions. Also, as the AOD is an integral quantity, it is not strongly influenced by changes of the vertical model structure.

13) Page 2 lines 1-2: radiative forcing (without s)

We corrected this.

14) Page 7: there is a section 2.3.1 without 2.3.2, could you check the numbering of subsections?

We removed the section number for "Dust emission tuning"

15) Figure 5 page 15: Long-dashed and short-dashed lines are difficult to distinguish. Could you improve it?

We changed the visualization of the dashed lines.